# Composition of Culturable Microorganisms in Dusts Collected from Sport Facilities in Finland during the COVID-19 Pandemic

**DOI:** 10.3390/pathogens12020339

**Published:** 2023-02-16

**Authors:** Maria (Aino) Andersson, Camilla Vornanen-Winqvist, Tuomas Koivisto, András Varga, Raimo Mikkola, László Kredics, Heidi Salonen

**Affiliations:** 1Department of Civil Engineering, School of Engineering, Aalto University, P.O. Box 12100, FI-00076 Aalto, Finland; 2Department of Microbiology, Faculty of Science and Informatics, University of Szeged, Közép Fasor 52, H-6726 Szeged, Hungary; 3International Laboratory for Air Quality and Health, Faculty of Science, School of Earth & Atmospheric Sciences, Queensland University of Technology, 2 George Street, Brisbane, QLD 4000, Australia

**Keywords:** *Aspergillus*, *Paecilomyces*, sport facilities, potential pathogens, airborne dust

## Abstract

Sport facilities represent extreme indoor environments due to intense cleaning and disinfection. The aim of this study was to describe the composition of the cultivated microbiota in dust samples collected in sport facilities during the COVID-19 pandemic. A dust sample is defined as the airborne dust sedimented on 0.02 m^2^ within 28 d. The results show that the microbial viable counts in samples of airborne dust (*n* = 9) collected from seven Finnish sport facilities during the pandemic contained a high proportion of pathogenic filamentous fungi and a low proportion of bacteria. The microbial viable counts were between 14 CFU and 189 CFU per dust sample. In seven samples from sport facilities, 20–85% of the microbial viable counts were fungi. Out of 123 fungal colonies, 47 colonies belonged to the potentially pathogenic sections of *Aspergillus* (Sections *Fumigati*, *Nigri*, and *Flavi*). Representatives of each section were identified as *Aspergillus fumigatus*, *A. flavus*, *A. niger* and *A. tubingensis.* Six colonies belonged to the genus *Paecilomyces.* In six samples of dust, a high proportion (50–100%) of the total fungal viable counts consisted of these potentially pathogenic fungi. A total of 70 isolates were considered less likely to be pathogenic, and were identified as *Aspergillus* section *Nidulantes*, *Chaetomium cochliodes* and *Penicillium* sp. In the rural (*n* = 2) and urban (*n* = 7) control dust samples, the microbial viable counts were >2000 CFU and between 44 CFU and 215 CFU, respectively, and consisted mainly of bacteria. The low proportion of bacteria and the high proportion of stress tolerant, potentially pathogenic fungi in the dust samples from sport facilities may reflect the influence of disinfection on microbial communities.

## 1. Introduction

During the first half of the 20th century, outdoor exercises were the main thrust of health-promoting activities, and outdoor air was considered to be healthy and therapeutic [1,2,3,4]. However, during the latter half of the 20th century, outdoor air pollution caused by vehicle traffic became difficult to avoid in urban environments. In 2013, the WHO declared outdoor urban air as a human carcinogen [1,5,6]. In recent decades, physical activities have moved into the built environment, and indoor athletics have become increasingly popular. Figure 1 illustrates the removal of football playing from outdoor to indoor environments. Nowadays, there are over 10,000 public and private commercial sport facilities, including sport halls, fitness centers and gyms, in Finland [7,8,9,10]. However, after the millennium shift, high hygiene levels and deprivation of exposure to diverse environmental microbiota in outdoor air have been associated with immune-mediated diseases [4]. Outdoor air in green environments has once again been indicated for maintaining health [3,4].

Physical activities are beneficial for health [8,9,10], but the indoor environment in sport facilities may also contribute to health risks not found in outdoor environments. As in most urban buildings, in sport facilities, mechanical ventilation provides indoor closed spaces with filtered inlet air. Filtering not only removes hazardous outdoor air pollutants, but may also reduce the beneficial components of outdoor air [4]. Specific indoor air quality (IAQ) problems identified in sport facilities are the accumulation of pollutants due to poor ventilation, exposure to high amounts of biocides associated with cleaning chemicals, emissions from plastic and rubber building materials, and contagious bacteria and pathogenic fungi [11,12,13,14]. Moreover, the COVID-19 pandemic has simultaneously increased the use of indoor disinfectants and the occurrence of fungal infections [15,16,17,18,19,20,21].

The risks of fungal and bacterial infections in sport facilities are widely recognized, but the diversity of pathogenic filamentous fungi in sport facilities in Finland is not well documented. The cultivable microbiota in indoor dusts collected during the pandemic have not been investigated. In this study, we show that the airborne dusts collected from Finnish sport facilities during the COVID-19 pandemic contained a high proportion of culturable pathogenic filamentous fungi, and a surprisingly low amount of culturable bacteria. This study is a continuation of the research presented earlier [9,22], but this research focuses on the proportion of cultivable fungi versus bacteria in settled indoor dusts from several Finnish sport facilities.

## 2. Results

### 2.1. Culturable Bacteria and Fungi in Samples of Sedimented Airborne Dusts Collected from Sport Facilities during the COVID-19 Pandemic

The culturable microbiota in nine airborne dust samples collected during the COVID-19 pandemic from seven sport facilities were investigated. Nine dust samples collected from urban and rural buildings were used as controls. The dust samples represented the amounts of airborne dust sedimented into three Petri plates within 2–4 weeks, or particles on 0.02 m^2^ per 14–28 d (Table 1). One of the rural control dust samples from a barn with bad quality hay was used as positive control for cultivation of mesophilic and thermotolerant fungi (able to grow at 37 °C). The other rural control dust sample, collected from an occasionally occupied and hardly cleaned summer house, was used as control for indoor microbiota not exposed to synthetic surfactants and disinfectants. The seven dust samples from urban “complaint-free” educational buildings collected before the pandemic were used as controls for random ordinary urban indoor microbiota.

Results in Table 1 visualize the total numbers of colonies identified as fungi or bacteria in each dust sample. In the dust samples from the sport facilities, the fungal and bacterial colony counts varied between no colony detected to 45 CFU, and 5 CFU to 144 CFU per dust sample, respectively. The fungal colony counts in the dust samples from sport facilities were >10 times higher than in the samples of urban indoor dust, but quite similar to the fungal colony counts in the samples of rural control dusts. The number of bacterial CFU in the dust samples from sport facilities was lower than the number of bacterial CFU in all the samples of control dusts, whereas the number of fungal CFUs in dust samples from sport facilities was largely similar to the number of fungal CFU in the control dust samples.

This indicates that the methods used for sampling, sample preparation and cultivation allowed growth of both bacteria and fungi, and enabled estimation of differences in microbial compositions of the dusts. The results also show that bacteria and fungi both grew on tryptic soy agar (TSA). On TSA, fungi able to grow at neutral pH competed successfully with the co-growing microbes. Very few bacterial colonies grew on malt extract agar (MEA).

### 2.2. Numbers of Dust Particles and Fungi versus Bacterial Colony Counts in the Sedimented Airborne Dusts

The numbers of particles sedimented onto the Petri plates during the sampling period were estimated and shown in Figure 2. The total microbial colony counts (CFU of bacteria + fungi) were related to the number of dust particles (particulate matter, PM), visible in microscope. The results in Figure 2 and Table 2 show that densities of PM and total microbial CFU varied between >100 PM and >100 CFU in dusts from SF1 and SF5 to <10 (PM) and 15 (CFU) in dust from SF3a. Both rural control dust samples contained high amounts of PM (>1000), and a high total microbial CFU (>1000). This indicated that the amounts of airborne dust that settled within the sampling period were very different between the different sport facilities and the rural control buildings.

The highest number of fungal colonies, 38 CFU and 45 CFU, was detected in dust samples from SF1 and SF5, respectively, which were the samples with the highest densities of PM. Notably, however, fungal colonies of 8 CFU and 15 CFU were also detected in dust samples from SF4 and SF6, respectively, which were samples containing small amounts of PM, <10. These dust samples contained a high proportion of viable fungi compared to the total microbial viable counts. In the rural control dust samples, the amounts of bacteria exceeded the amounts of fungi by >10 times.

The results in Table 2 show that the fungal viable counts in most of the dust samples from sport facilities (8 out of 9 dust samples) were between 13% and 85% of the total microbial viable counts, whereas the fungal colony counts were <1% of the total microbial viable counts in the control samples of rural dusts. In the control samples of urban dusts, the fungal viable counts were between <1% and 5% of the total viable counts. The microbial composition in the dust samples from the sport facilities also differed from each other. The results also revealed a low number of total microorganisms and a high proportion of fungi in two sport facility dusts, SF4 and SF6, exhibiting a low density of PM.

### 2.3. Tracking Diversity of Fungal Colonies from the Samples of Settled Dusts Collected from Sport Facilities—Identification of the Isolates to Genus, Section and Species Levels

Fungal colonies (*n* = 123) representing different colony morphology and different microscopic morphology were purely cultured and screened for pathogenic potential by their ability to grow at 37 °C [22]. The 53 potentially pathogenic isolates found were identified to the genus, section and species levels by classical morphological criteria; 11 isolates were identified to species level using molecular methods. A flow chart of the results is presented in Figure 3. Out of the 70 less likely pathogenic isolates (those not growing at 37 °C), only six strains were identified. From the control dust samples collected from a barn and a summer house, 30 out of 217 isolates were identified to section level and considered potentially pathogenic (Figure 4). The results presented in Table 3 show that the fungal colonies from six dust samples collected from the sport facilities consisted mainly of potentially pathogenic isolates belonging to the genera *Aspergillus* and *Paecilomyces*, respectively. The majority of the fungal colonies in the rural control dust samples, CD1 and CD2, were the less likely pathogenic *Aspergillus* and *Penicillium* isolates not growing at 37 °C. From the urban control dust samples, only five fungal colonies tested positive for growth at 37 °C, and were identified by microscopy as yeasts. 

Specifically, the results in Table 3 show that out of the nine tested dust samples from sport facilities, six dust samples contained strains of potentially pathogenic fungal species belonging to *Aspergillus* sections *Flavi*, *Fumigati* and *Nigri*. Four dust samples contained potentially pathogenic strains of *Paecilomyces* sp. Two dust samples contained a majority of colonies that tested negative for pathogenic potential. Most of these colonies remained unidentified, but some isolates of *Penicillium* sp., *Aspergillus* section *Nidulantes*, and *Chaetomium cochliodes* were identified. The majority of the isolates from the rural control dust samples represented nonpathogenic species belonging to *Aspergillus* sections *Nidulantes*, *Circumdati* and the genera *Trichoderma* and *Penicillium.* The potentially pathogenic colonies from the rural control dust sample CD1 represented *Aspergillus* sections *Fumigati* and *Nigri.* The rural control dust sample CD2 from an uncleaned summer house and one sport facility dust SF2a contained no fungal isolates that tested positive for pathogenic potential. No colonies of potentially pathogenic filamentous fungi were detected from the seven urban control dust samples.

Summarizing the results in Table 1, Table 2 and Table 3, the dust samples collected during the COVID-19 pandemic from the sport facilities contained microbial colony counts that were 100 times smaller than those of the rural control samples from a hay barn and a summer house. However, the dust samples from the sport facilities differed from the control samples by a higher proportion of fungi in the viable counts for total microorganisms, and six of the samples contained a higher proportion of potentially pathogenic fungi, (81%; range: 50–100%), in the viable counts for total fungi. The two control dusts contained 18% and <1%, of potentially pathogenic fungi in the total fungal viable count. The dust samples from the urban control buildings contained no colonies identified as potentially pathogenic fungi.

### 2.4. Numbers of Persons Visiting the Different Locations, Density of Particles and Microbial Colony Counts in the Settled Dusts

The number of visiting persons was restricted in the sport facilities during the pandemic. The results summarized in Table 4 show that there seemed to be a positive connection between the numbers of visiting persons in the locations, the density of dust particles sedimented into the sampling devices, and the bacterial colony counts of the dusts. However, a high proportion of pathogenic fungi was recorded from facilities with low amounts of visiting persons, low densities of particles in the settled dusts and low bacterial colony counts.

### 2.5. Summary of Results

Summarizing the results in Table 1, Table 2, Table 3 and Table 4, the dust samples from the seven sport facilities contained lower microbial colony counts compared to control dusts from urban educational buildings and those from a barn and a summer house. However, the dust samples from the sport facilities contained a high proportion of fungi in the colony counts for total microorganisms. Six of the samples contained a high proportion of potentially pathogenic species belonging to the *Aspergillus* sections *Flavi*, *Fumigati* and *Nigri*, and representants of the genus *Paecilomyces*, comprising 50–100% of the total fungal colony counts. The corresponding percentages for the control dust samples were only 18% (sample of barn dust) and <1% (sample of summer house dust). The proportion of 100% of potentially pathogenic fungi in the total fungal colony counts was recorded in two dusts collected from facilities with low numbers of visiting persons, low numbers of sedimented dust particles and low bacterial colony counts.

## 3. Discussion

Sport facilities may provide hostile environments for microorganisms because of the effective and regular cleaning and disinfection [23,24]. During the COVID-19 pandemic, disinfection of public buildings was globally intensified [16,17,25,26,27]. This article describes the cultivable microbiota in samples of sedimented dusts collected from sport facilities during the COVID-19 pandemic. The buildings represented ordinary low risk indoor spaces; no serious indoor air quality complaints or water damage were reported. To our knowledge, this is the first report of potentially pathogenic fungi as a large proportion (but not a high number) of the cultivable microbiota of dust samples from sport facilities.

In this study, we used passive collection of settled dust, a sampling method used in several studies characterizing microbial communities in different indoor environments based on analysis of extracted microbial DNA [28,29,30,31]. We hypothesized that this sampling method would favor stress-tolerant and cleaning chemical-resistant microbes as potentially pathogenic *Aspergillus* spp. [22,32,33]. We chose this dust sampling method for comparing the stress-tolerant culturable microbiota in sedimented dusts from different indoor environments; indoor spaces were subjected to intense cleaning (sport facilities during the pandemic), ordinary cleaning (schools before the pandemic), and an absence of cleaning (a hay barn and a summer house).

The composition of the cultivable microbiota in nine dust samples from sport facilities differed from the compositions in the dust samples from the schools and from the rural buildings: (1) In five dust samples, the fungal colony counts were high compared to the total microbial viable counts (≥20%); (2) in six dust samples, a high proportion (>50%) of the fungal viable counts consisted of potentially pathogenic fungi; and (3) the bacterial viable counts in sport facility dusts were low, and >100 times lower than those in the rural control dusts collected from the buildings never cleaned or disinfected. The low bacterial viable counts coincided with the high proportion of pathogenic fungi. In the urban control dusts (samples of school dust) collected before the pandemic, the proportion of viable fungi was ≤5% of the total microbial viable counts.

The majority of the metabolically active building mycobiota are cultivable and identifiable to the genus level by cultivation-based conventional methods [34]. Non-selective media increase microbial diversity in the obtained viable counts compared to selective media. Complex nonselective culture media without sugars such as TSA (pH 7.2) restrict the growth of rapidly growing microorganisms [35]. The TSA medium also allowed growth of common pathogenic fungi [22].

The virulence of pathogenic fungi is connected to viability and ability to sense and cope with environmental stress in hostile environments. The virulence factors required for invasive fungal infections in humans include the ability to grow at the pH of human tissues, 7.2–7.4, at ≥37 °C [36,37,38]. Strains of the potentially pathogenic species *Trichoderma longibrachiatum* and the less likely pathogenic *Aspergillus westerdijkiae* were isolated from viable indoor samples counted on TSA [39,40]. Additional virulence factors of *Aspergillus* spp., heat resistance and adaptation to extreme pH values, have been shown to be connected to their resistance to disinfectants, cleaning chemicals and antifungal drugs [22,32,33,36,37,38,41]. The explanation for the high proportion of potentially pathogenic fungi in sport facility dusts may be that the intense disinfection in sport facilities irradicated most of the viable microorganisms, but not the stress-tolerant and virulent potentially pathogenic fungi.

Screening for pathogenic potential (growth at 37 °C at neutral pH), and characterization to the genus/section level seemed to be successful methods for the detection and enumeration of the viable potentially pathogenic fungi in the collected dust samples. Identification to the species level for evaluation of risk group and biohazard levels required DNA sequence analysis of marker genes [34,42,43]. Results in Figure 3 and Table 5 show that dust samples from sport facilities contained strains of *A. flavus* and *A. fumigatus* classified as risk group 2 (RG-2) and bio-safety level 2 (BSL-2) organisms, species known to cause human infections [22,42]. Recently, the WHO ranked fungal pathogens into three priority groups (critical, high, and medium). *A. fumigatus* was the only representant of the genus *Aspergillus* ranked into the critical priority group [44]. *A. niger* and *A. tubingensis* are classified as risk group 1 and 2 organisms, respectively, although human infections have been reported [42,43]. Out of the nine control dust samples, the only one which contained strains of *Aspergillus* section *Fumigati* and section *Nigri* was the dust sample from the barn containing moldy hay. This indicates that *Aspergillus* spp. belonging to RG-2 are not ubiquitous in Finnish buildings. The two potentially pathogenic *Paecilomyces* sp. were found to be close to, but not identical to *Paecilomyces variotii*. These strains possibly represent a new, as-yet undescribed species and could not be assigned to any risk group. Nevertheless, *P. variotii* is classified as an RG-2 organism [42,43].

The airborne bacteria and fungi measured with an RCS microbial air sampler revealed that the ratios of total bacteria and total fungi were lower indoors than outdoors in all sport facilities [9]. It could be speculated that users might carry fungal spores from outdoor air into sport facilities. The sport facilities were all mechanically ventilated. Uncleaned unsubstituted inlet filters of air conditioning and mechanical ventilation might have served as fungal reservoirs in hospitals [45,46].

In indoor air, impurities are inflated compared to outdoor air wherein the content of every breath differs from the next [4,47]. In forested and green environments, the air contains a diverse microbiota. Environmental microbial exposure, human commensal microbiota, and immunological pathways are generally assumed to be interconnected. A high hygiene level and urban lifestyle may result in microbial imbalance, referred to as dysbiosis, which has been associated with immune-mediated diseases [4,48,49,50,51].

## 4. Conclusions

As the number of tested dust samples is too low to be able to draw far-reaching conclusions, this study is merely to be considered a pilot work concerning differences in microbial communities in indoor dusts from rural and urban buildings, and from buildings subjected to intense cleaning and disinfection. The viable potentially pathogenic fungi detected meant hardly any health risk for the users. The low bacterial viable counts in the dusts from the sport facilities compared to the control dusts may reflect the use of disinfectants, the low numbers of users and filtration of the inlet air. The microbial composition in the sedimented airborne dust in sport facilities during the COVID-19 pandemic may illustrate some unexpected and unwanted effects of intense cleaning and disinfection of indoor closed spaces.

## 5. Materials and Methods

### 5.1. Sampling Design

To investigate the composition of the cultivable microbiota in settled indoor dusts from sport facilities and from urban and rural control buildings, dust collection and cultivation of microbes were designed as shown in Figure 5. Nine dusts from 7 sport facilities were compared to the 9 control dusts, themselves consisting of 7 dusts from 3 urban educational buildings and 2 dusts from 2 rural buildings. The control dusts were collected from urban educational buildings which were complaint-free concerning IAQ, from a barn containing moldy hay, and from a summer house never cleaned and disinfected. The dust samples were investigated for colony counts (CFU) of the following: (a) total microorganisms (bacteria + fungi); (b) total fungi (fungi growing at 24 °C + fungi growing at 37 °C); and (c) fungal isolates with pathogenic potential screened by the ability to grow on TSA (pH 7.2) at 37 °C. Potentially pathogenic isolates were identified to genus, section and species level. The proportion of viable counts of potentially pathogenic fungi in the microbiota was related to the viable counts of total fungi, of total microorganisms, and of bacteria.

### 5.2. The Study Buildings

The seven investigated sport facilities are located in 3 cities in southern Finland. The 5 reference buildings were three educational buildings in the 3 cities, a hay barn in a village in southern Finland, and a summer house in a village in central Finland. Pictures illustrating the three types of buildings are shown in Figure 6. Details of the sampling sites are presented in Table 6.

### 5.3. Collection of Sedimented Airborne Dust by a Standardised Method

Passive collection of airborne dust settling into Petri plates over a specified time period is a method of detecting differences in amounts of aerosolized dust particles, and also enables the detection of differences in the composition of dust-borne microbiota. Passive collections of settled dust have been used in several studies wherein the microbial communities in different indoor environments have been investigated, characterized and compared [52,53,54,55].

From the sport facilities and the control buildings, dusts were sedimented for 2–4 weeks into sterile polystyrene Petri plates (Berner, 90 × 15 mm, SILEÄ0099/S). The plates were placed 150 cm to 220 cm above floor level, and as far as possible from the inlet and outlet ventilation ducts, doors, windows, and heating sources. A 50 cm minimum air column was ensured above the plates to allow free air movement. After sampling, the plates were sealed with parafilm and stored at room temperature. The dusts from the sport facilities were analyzed at 2–4 months after collection, and the reference building dust samples 2–6 months after collection.

The use of the sport facilities was strongly restricted during the sampling period due to the ongoing COVID-19 pandemic. Largely, only professional athletes and teams and school classes were permitted to use the facilities. Therefore, the occupancy rate was significantly lower than that during normal use, and the occupancy varied depending on the sampling site. The estimated occupancy rate in each sampling site during the sampling period is shown in Table 5.

### 5.4. Cultivation of Dust Samples, Enumeration of Bacterial and Fungal Colonies, Isolation of Pure Cultures and Testing for Pathogenic Potential

From each site, 3 Petri plates of sedimented dust were collected. The dusts were removed from the plates by suspending in 0.7 mL PBS to flush the bottom surface of the plate. From each sample, 2.1 mL PBS + 0.05% of Tween (Sigma Aldrich, St. Louis, MO, USA) was collected. The bottoms of the plates were then wiped with a moist cotton swab, and the swab was then immersed in the collected PBS. Ten µL from each sample of the visibly cloudy PBS was inspected by microscopy (400× magnification; Olympus CKX41, Tokyo, Japan) and an image recording software (cellSens^®^ standard v. 11.0.06, 2012, Olympus Soft Imaging Solutions GmbH, Münster, Germany) to enumerate the dust particles removed from the plastic surface of the plates. Of the collected PBS, 100 µL was applied and diluted on cultivation plates, and incubated according to the schedule shown in Figure 7.

The dusts were cultivated on malt extract agar (MEA), pH 5.4, and tryptic soy agar (TSA), pH 7.2 (Oxoid Ltd., Thermo Fisher, Heysham, Ireland). The plates were sealed with gas-permeable tape before incubation. To detect *Trichoderma* isolates growing as mycoparasites, the plates incubated at room temperature were inspected for the final time after 6 weeks of incubation.

### 5.5. Tracking Diversity of Fungal Isolates, Assignation to Genera and Sections, Testing for Pathogenic Potential, and Identification to the Species Level

The fungal colonies cultured pure on MEA at 22 °C were screened for pathogenic potential by their growth at 37 °C on TSA (pH 7.2) [22,30,31,32,33]. Some isolates identified as yeast by the microscope were not further identified. All isolates of filamentous fungi that screened positive for pathogenic potential were identified to genus and section level as *Aspergillus* sections *Flavi*, *Fumigati*, *Nigri* and *Paecilomyces* sp. by colony morphology on MEA, size and morphology of conidiophores and conidia in a light microscope (400× magnification), with the equipment described above [22,52]. The identification of the isolates was confirmed by comparison to selected strains isolated from the sport facilities and identified earlier [22,52]. Most of the selected strains listed in Table 6 were identified to the species level in previous studies by sequence analysis of *cmd*A [53] and/or *rbp2* [54]. The identified strains, the gene loci used for identification and the GenBank accession numbers are listed in Table 6. The two *Paecilomyces* strains in Table 6 could not be assigned to an existing species based on the *cmd*A locus amplified during this study with primers Cmd5 and Cmd6, according to [55].

The strains that tested negative for pathogenic potential were identified to the species level as *Chaetomium cochliodes,* as shown in (Table 6). Strains belonging to *Aspergillus* section *Nidulantes* and *Aspergillus westerdijkiae* were identified by comparison to strains SL/3 and PP2, respectively, identified by DSMZ [22] and according to Samson et al. [52]. *Trichoderma* and *Penicillium* strains were identified to the genus level based on colony morphology and the morphology of conidiophores, by comparison to reference strains described [56,57] and according to Samson et al. [52]. The morphology of the isolates identified is visualized in Appendix A.

## Figures and Tables

**Figure 1 pathogens-12-00339-f001:**
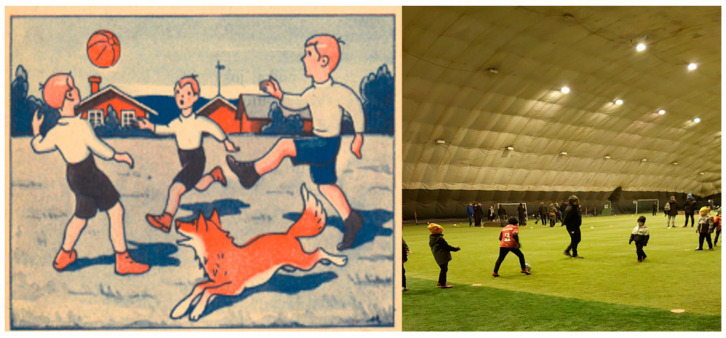
Football then and now. This figure illustrates the movement of physical activities indoors. Left panel: Depiction of health education in Finland from the first half of the 20th century, which stressed the benefits of outdoor exercises for the prevention of tuberculosis [3]. Right panel: Modern football players inside a built environment, possibly avoiding traffic air pollution in urban air and the effects of the outdoor climate (snow and rain) in Finland (photo: Sanna Sulopuisto).

**Figure 2 pathogens-12-00339-f002:**
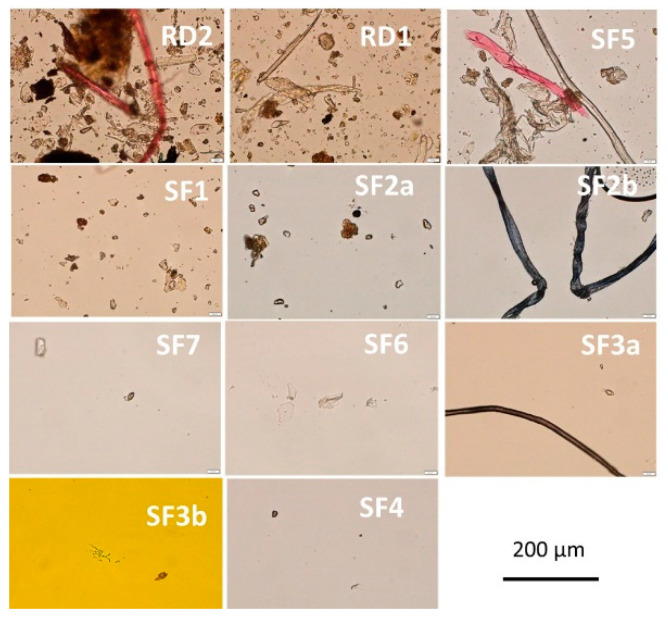
Densities of particles of 11 settled dusts captured in light microscope (400×). Airborne dust samples SF1-SF7 were collected by sedimentation into Petri plates from sport facilities. The reference dusts RD1 and RD2 were collected from a hay barn and an old summer house. Each dust sample (collected in three fallout plates) was suspended in 2 mL phosphate-buffered saline (PBS). Some 10 µL of the suspension was examined in five microscopic fields. The view shown in each picture is representative of the recorded views in the five examined microscopic fields, thus representing an average view of each dust.

**Figure 3 pathogens-12-00339-f003:**
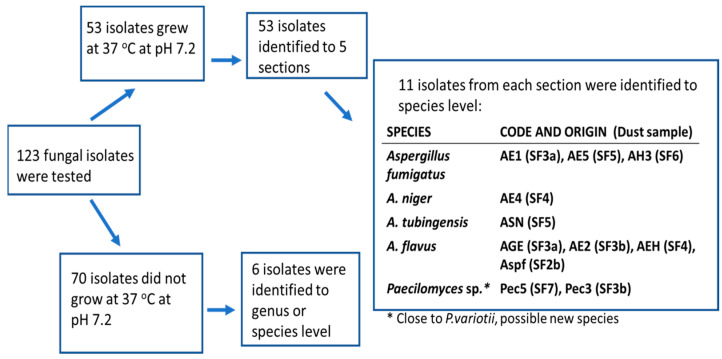
Identification of the isolates from dust samples collected from sport facilities considered as potentially pathogenic based on their ability to grow at 37 °C and neutral pH. From each section and genus, selected strains from different samples were identified using molecular methods. The codes for the dust samples from the different sport facilities are shown in parentheses (SF2-SF7).

**Figure 4 pathogens-12-00339-f004:**
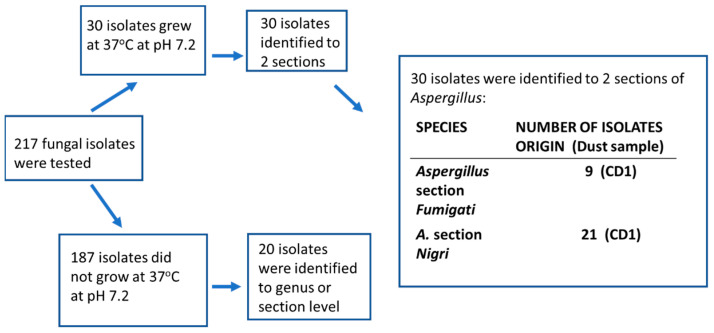
Identification of the isolates from the rural control dust samples, a hay barn (CD1) and a summer house (CD2). The 30 isolates growing at 37 °C were from the barn; the summer house contained no isolates growing at 37 °C.

**Figure 5 pathogens-12-00339-f005:**
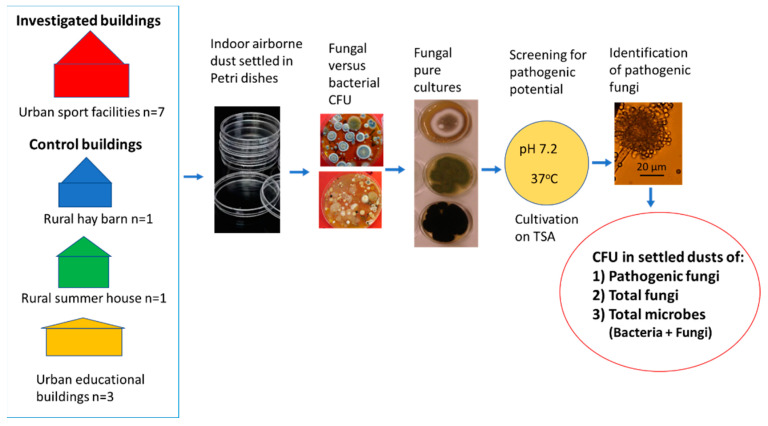
The experimental design for comparison of the fungal and bacterial colony counts in settled air-borne dusts and for screening for potentially pathogenic fungi. Pathogenic potential was indicated by growth on tryptic soy agar (TSA) at 37 °C.

**Figure 6 pathogens-12-00339-f006:**
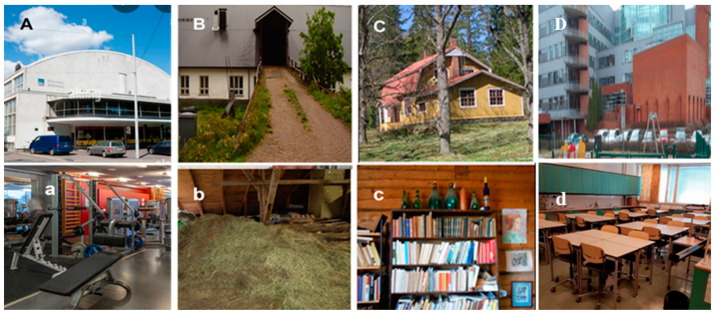
The outside (panels **A**–**C**) and inside (panels **a**–**c**) views of the urban and rural buildings examined in this study. Panels A and a represent views of an urban sport facility in Helsinki. The further panels show the two rural reference buildings, a hay barn (**B**,**b**), collection site for dust CD1, and a rural summer house (**C**,**c**), collection site for dust CD2. (Panels **D**,**d**) shows views of an urban educational building in Espoo.

**Figure 7 pathogens-12-00339-f007:**
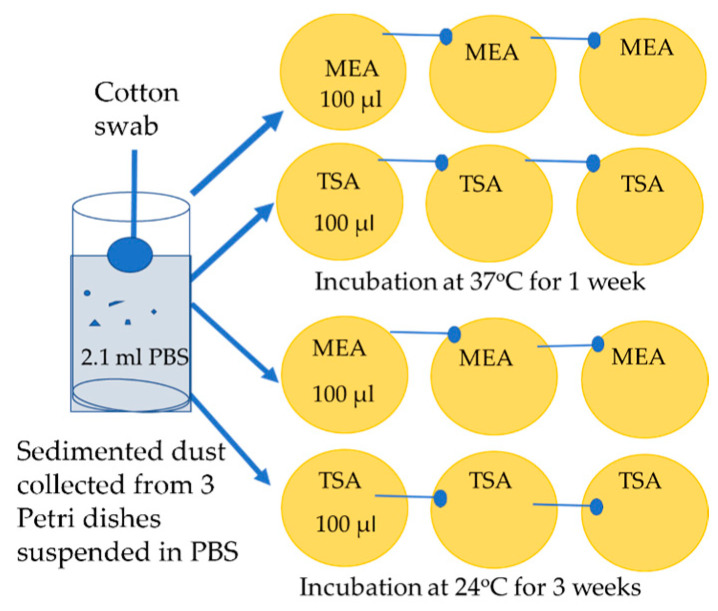
Cultivation of a sample of sedimented dust. The dust collected from each sample was suspended in 2.1 mL PBS, and 100 µL was applied on the first vertical row of plates. The liquid was spread with the cotton swab on the first plate, the next two plates were wiped with the same swab.

**Table 1 pathogens-12-00339-t001:** Cultivable bacteria and fungi grown on malt extract agar (MEA, pH 5.5) and tryptic soy agar (TSA, pH 7.2) at 24 and 37 °C. SF1-SF7 = dust samples from sport facilities, SD1-SD3= samples of school dusts, CD1-CD2 = samples of rural control dusts.

Sites	Total Bacteria	Total Fungi	Microbial Colony Count CFU of Fungi, CFU of Bacteria (in Brackets)			
Dusts	CFU	CFU	MEA 24 °C	TSA 24 °C	MEA 37 °C	TSA 37 °C
Dusts from sport facilities sedimented for 4 weeks during the COVID-19 pandemic (June–July 2021)						
SF1	85	38	31 (ND)	5 (25)	2 (ND)	ND (60)
SF2a	54	<1	ND (ND)	ND (33)	ND (ND)	ND (24)
SF2b	13	2	2 (ND)	ND (2)	ND (ND)	ND (11)
SF3a	9	6	2 (ND)	2 (3)	2 (3)	1 (3)
SF3b	14	4	ND (ND)	3 (14)	1 (ND)	ND (ND)
SF4	5	15	8 (ND)	4 (4)	2 (ND)	1 (1)
SF5	144	45	5 (1)	30 (79)	1 (ND)	9 (65)
SF6	5	8	3 (ND)	ND (ND)	3 (ND)	2 (5)
SF7	23	4	ND (ND)	ND (15)	4 (ND)	ND (8)
Reference sites						
Urban dusts sedimented for 2 weeks in public buildings before the pandemic (December 2018–April 2019)						
SD1a	192	ND	ND (ND)	ND (116)	ND (ND)	ND (76)
SD1b	223	ND	1(ND)	ND (120)	ND (ND)	ND (103)
SD2a	94	1	1 (ND)	ND (57)	ND (3)	ND (34)
SD2b	132	1	1 (ND)	ND (87)	ND (ND)	ND (45)
SD3a	205	10	10 (ND)	ND (110)	ND (ND)	ND (95)
SD3b	184	7	4(ND)	2 (80)	1 (ND)	ND (104)
SD3c	42	2	1 (ND)	ND (8)	1 (ND)	ND (34)
Rural dust from a hay barn with moldy hay collected for 4 weeks during the pandemic in November 2021						
CD1	>1000	191	161 (ND)	ND (>500)	30 (ND)	ND (>500)
Rural dust from a summer house collected for 4 weeks during the pandemic in November 2021						
CD2	>100	26	3 (ND)	23 (>100)	ND (2)	ND (>100)

ND = No colonies detected.

**Table 2 pathogens-12-00339-t002:** Density of particles visible in microscope (PM) and microbial colony counts (CFU) in sedimented dust samples collected from sport facilities (SFs) in Finland.

Location Dust Sample	Numbers of PM in Samples	Total Microbial CFU in Sample	Fungal CFU in Sample	Proportion of Fungal CFU in Total Microbial CFU (%)
Dust samples from sport facilities sedimented for 4 weeks during the COVID-19 pandemic (June–July 2021)				
SF1	>100	123	38	31
SF2a	>30	54	ND	ND
SF2b	<10	15	2	13
SF3a	<10	15	6	40
SF3b	<10	20	4	20
SF4	<10	20	15	85
SF5	>100	189	45	23
SF6	<10	14	8	62
SF7	<10	27	4	15
Reference sites				
Urban dust samples sedimented for 2 weeks from public buildings before the pandemic (December 2018–April 2019)				
SD1a	NI	192	ND	<1
SD1b	NI	223	ND	<1
SD2a	NI	95	1	1
SD2b	NI	133	1	<1
SD3a	NI	215	10	4
SD3b	NI	191	7	4
SD3c	NI	44	2	5
Rural dust samples from a hay barn with moldy hay sedimented for 4 weeks during the pandemic (November 2021)				
CD1	>1000	>2000	191	<10
Rural dust from a summer house sedimented for 4 weeks during pandemic (November 2021)				
CD2	>1000	>1000	26	<1

ND = No colonies detected. NI = Not investigated.

**Table 3 pathogens-12-00339-t003:** Species diversity and pathogenic potential of colonies of filamentous fungi isolated from sedimented dusts collected in sport facilities in Finland. Pathogenic potential was indicated by growth at 37 °C at neutral pH.

Location	Total CFU in Sample					
Dust Sample	Bacteria	Fungi	Genus/Section	CFU in Sample	Growth at 37 °C at pH 7.2	Potentially Pathogenic CFU in Total Fungal CFU (%)
SF1	85	38	*Aspergillus* section *Nigri*	2	+	
			*Aspergillus* section *Nidulantes*	3	-	
			*Aspergillus* section *Flavi*	1	+	5
			*Chaetomium cochliodes*	2	-	
			Unidentified colonies including *Penicillium* sp.	30	-	
SF2a	54	ND	ND	ND	ND	
SF2b	13	6	*Aspergillus* section *Flavi*	2	+	100
			Yeasts	4	+	
SF3a	9	6	*Aspergillus* section *Flavi*	2	+	
			*Aspegillus* section *Fumigati*	1	+	100
			*Aspergillus* section *Nigri*	2	+	
			*Paecilomyces* sp.	1	+	
SF3b	14	4	*Aspergillus* section *Flavi*	1	+	
			*Paecilomyces* sp.	1	+	50
			Unidentified colonies	2	-	
SF4	5	15	*Aspergillus* section *Fumigati*	4	+	
			*Aspergillus* section *Flavi*	7	+	100
			*Aspergillus* section *Nigri*	3	+	
SF5	144	45	*Aspergillus* section *Flavi*	4	+	
			*Aspergillus* section *Nigri*	6	+	22
			Unidentified colonies including *Penicillium* sp.	37	-	
SF6	5	9	*Aspergillus* section *Fumigati*	3	+	
			*Aspergillus* section *Nigri*	2	+	88
			*Paecilomyces* sp.	3	+	
			*Penicillium* sp.	1	-	
SF7	23	4	*Aspergillus* section *Flavi*	3	+	
			*Paecilomyces* sp.	1	+	100
Control dusts samples						
CD1	>500	191	*Aspergillus* section *Nigri*	21	+	
			*Aspergillus* section *Fumigati*	9	+	
			*Aspergillus* section *Circumdati*	3	-	18
			*Trichoderma* sp.	1	-	
			Unidentified colonies including *Penicillium* sp.	157	-	
CD2	>100	26	*Penicillium* sp.	16	-	<1
			*Aspergillus* section *Nidulantes*	10	-	

ND = no colonies detected. Pathogenic potential was indicated by ability to grow at neutral pH at 37 °C.

**Table 4 pathogens-12-00339-t004:** Number of visiting persons in the sport facilities during the sampling period of settled airborne dusts, the density of dust particles visible in microscope (PM) and the microbial colony counts (CFU) in the corresponding dust samples.

Sport Facility		Dust Samples Sedimented for 4 Weeks				
Location	No. of Persons	Particles PM	Total Bacteria CFU	Total Fungi CFU	Pathogenic Fungi CFU	Pathogenic Fungi CFU in Total Fungal CFU (%)
Sport facilities						
SF1	~5000	>100	85	38	2	5
SF2a	~4000	>30	54	ND	ND	
SF2b	<50	<10	13	6	6	100
SF3a	~1000	<10	3	6	6	100
SF3b	Closed	<10	14	4	2	50
SF4	~300	<10	5	15	15	100
SF5	~5000	>100	144	45	10	22
SF6	<50	<10	5	8	8	88
SF7	<500	<10	23	4	4	100
Control sites						
CD1	<2	>1000	>500	191	34	18
CD2	<2	>1000	>500	26	ND	<1

ND = no colonies detected.

**Table 5 pathogens-12-00339-t005:** Selected strains representing the potentially pathogenic genera identified to species level.

Section	Species	Strain ID	Location of Isolation	GenBank Accession Number		
				*rpb2*	*cmdA*	Reference
*Fumigati*	*Aspergillus fumigatus*	AE1	SF3a		OP295388	[22]
	*Aspergillus fumigatus*	AE5	SF5		OP295389	[22]
	*Aspergillus fumigatus*	AH3	SF6		OP295390	[22]
*Nigri*	*Aspergillus niger*	AE4	SF4		OP295391	[22]
	*Aspergillus tubingensis*	ASN	SF5		OP295393	[22]
*Flavi*	*Aspergillus flavus*	AGE	SF3a		OP295387	[22]
	*Aspergillus flavus*	AE2	SF3b		OP295385	[22]
	*Aspergillus flavus*	AEH	SF4		OP295386	[22]
	*Aspergillus flavus*	ASpf	SF2b		OP356696	[22]
	*Paecilomyces* sp.	Pec5	SF7		OP356695	Recent study
	*Paecilomyces* sp.	Pec3	SF3b		OP356694	Recent study
	*Chaetomium cochliodes*	CH2	SF1	OP356691	OP295395	[22]
	*Chaetomium cochliodes*	CH3	SF1	OP356692		[22]

**Table 6 pathogens-12-00339-t006:** Locations, activities and occupancy characterizing the sampling sites of urban sport facilities and two rural reference buildings. The sport facilities were built between 1935–2008, had mechanical ventilation and were under modest use during the sampling time [3]. The reference buildings were >100 years old, had natural ventilation and were only sporadically visited by humans.

Dust Sample Site	Location	Building	Occupancy (Number of Persons) in the Sampling Site during the Sampling Period	Sampling Site	Building Described in Ref [3] Code
SF1	Vantaa	A bedrock shelter used as sport hall	~500	Athletics area	11
SF2a	Helsinki	Sport hall	~4000	Athletics area	7
SF2b	Helsinki		<50	Gym	7
SF3a	Espoo	Sport hall	~900–1000	Ball game area	2
SF3b	Espoo		(Closed during sampling)	Gym	2
SF4	Espoo	A bedrock shelter used as sport hall	~300	Table tennis area	5
SF5	Helsinki	Sport hall	~5000	Gymnastics area	6
SF6	Helsinki	Sport hall	<50	Gym	8
SF7	Vantaa	Sport hall	<500	Gym	13
SD1a	Vantaa	School	20–30	Classroom	
SD1b	Vantaa	School	20–30	Classroom	
SD2a	Espoo	School	20–30	Classroom	
SD2b	Espoo	School	20–30	Classroom	
SD3a	Helsinki	Day care	20–30	Playroom	
SD3b	Helsinki	Day care	20–30	Playroom	
SD3c	Helsinki	Day care	20–30	Playroom	
CD1	Hautjärvi	Hay barn (store of discharged mouldy hay)	<2	Hay barn	
CD2	Joroinen	Summer house (not cleaned since 1947)	<2	Summer house	

The buildings coded SF were described and coded as 11,7,2,5,6,8,13 by Koivisto [3].

## Data Availability

Raw data are available upon request to the authors.

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
