# Peer review of "Composition of Culturable Microorganisms in Dusts Collected from Sport Facilities in Finland during the COVID-19 Pandemic"

_pathogens, 2023, doi:10.3390/pathogens12020339_

Round 1

Reviewer 1 Report (Previous Reviewer 1)

The work is mow improved.

Author Response

We found no questions or comments from Reviewer I,

Reviewer 2 Report (Previous Reviewer 2)

This study  focuses on the proportion of cultivable fungi versus bacteria in settled indoor dusts from several Finnish sport facilities. need some improvements before publication:

- Include goal in the abstract

- This study cannot corroborate this sentence: ...but it does not replace the benefits of exercising outdoors in environments 37 providing exposure to diverse environmental microbiota. I suggest to be removed from Conclusions section.

- Line 87: "Nine dusts" should be replaced by "nine samples of dust". This should be revised thorugh all the MS

- The paragraph starting in line 105 is confused. MEA has cloranfenicol so only bacteria resistante to that antibiotic will grow. TSA is not proper to culture fungi, but only total bacteria. Thus, conclusions regarding this information shouldn´t to be stated.

- Section 2.3 genus should be first mentioned intead of species

- the 123 isolates are coming from wich samples? I suggest a flow chart from this section

- Avoid taxonomic diferences in the nomenclature of fungi. You should keep the section level through all the MS (even in the abstract)

- Why not all the isolates were identified?

- Table 4 needs some statistical analyses

- Remove this sentence" To our knowledge, this is the first report of potentially 233 pathogenic fungi as a large proportion of the cultivable microbiota from urban buildings 234 not connected to water damages or IAQ complaints." This is a very common situation in sevral European countries.

- The use of the media without antibioticts for fungi growth will impact their growth due the bacteria that can thrive and this should be discussed and not presented has only a feature

Author Response

Answere to Rev.II is attached

Reviewer 3 Report (Previous Reviewer 3)

The paper looks a little bit better, but still in my opinion, it is not worth publishing and will not be because of the substantial methodological flaws. The authors call the results obtained "striking", "interesting", and exceptional", but they are rather poor and ordinary. The authors tried to justify the use of their sampling methodology by referring to [56], for example, but in this study, the passive collection of dust samples was used for further DNA extraction, while in the current study, it was employed for further cultivation, when the storage of samples for 2-4 months before processing, and their cultivation in a complicated several-stepped way might certainly lead to the loss of fungal and bacterial diversity. Moreover, the sampling was conducted only ones and the absence of temporal repetitions also decreases the result quality. In my strong opinion, adding the three educational buildings as reference buildings where the dust collection was conducted not in the same time (before the pandemic) is artificial and does not allow the relevant comparison.

The subsection 5.5. vaguely describes the identification processes, with unnecessary details. By the way, a strain must be identified not by "comparison to reference strains described" (line 467), but by comparison with the descriptions in relevant identification books.

All other corrections and suggestions are inserted into the PDF file of the manuscript, which is attached.   

Author Response

Answere to Rev.III attached

Round 2

Reviewer 2 Report (Previous Reviewer 2)

The authors address all the comments raised.

Author Response

the answers are attached below

Reviewer 3 Report (Previous Reviewer 3)

My opinion have not been changed - the paper is not worth publishing, and will not be because of the substantial methodological flaws:

-          indirect several-stepped procedure of the isolation of microorganisms from dust samples, which might certainly lead to the loss of viable fungal and bacterial diversity. The authors tried to justify the use of their sampling methodology by referring to [56-58], but in these studies, the passive collection of dust samples was used for further DNA extraction and not for the determination of viable bacteria and fungi.

-          the absence of temporal repetitions;

-          the absence of relevant control sampling. In my strong opinion, neither the chosen rural buildings, no the public buildings sampled before the pandemic can serve as proper controls.

Abstract consists of near 400(!) words and contains a lot of plain number information unsuitable for this section.

Tables are unnecessarily numerous and contain the repeating information, while a very simple, but important and useful datum - the overall number of fungal species isolated from the dust samples, is just absent. Moreover, the tables are built in a rather strange way, embedding the information which should be placed either in the captions or footnotes, like "Dust samples from sport facilities sedimented for 4 weeks during the Covid-19 pandemic (June-July 2021)".

Once again, Discussion contains commonplaces and the statements, which are hardly associated with the study topic. The authors must take into account that many non-pathogenic fungi can grow also at neutral pH (line 284). And the statement, that "the non-selective cultivation media used in this study, MEA and TSA, without fungicides or antibiotics have proved successful for detection of new fungal species" (lines 286-287) is more than strange, because many new species of soil fungi were isolated on cultivation media containing antibiotics (than being grown in pure cultures probably without antibiotic).  

The authors are right concluding that "the number of tested dust samples is too low to be able to draw far-reaching conclusions". The abovementioned flaws in sampling design and methodology can also hardly support the conclusions on 'unusual" or "exceptional"(!) microbial composition in the studied dust samples. Probably, this "pilot work" may be considered for publication as a short communication after its substantial revision, but as a full-length research paper – definitely not.  

Author Response

Answers to reviewer 3:

The discussion is rewritten

The abstract is shortened

The text has checked for edition and language

This manuscript is a resubmission of an earlier submission. The following is a list of the peer review reports and author responses from that submission.

Round 1

Reviewer 1 Report

I suggest authors to improve Material and Methods section and afterwards to change Results Section accordingly. Some specific points:

Material and Methods

Line 277. I do not agree that fungal growth 37°C lead to identification of pathogen morphotypes. There are other tests (growth on Blood agar, experiments with Zebra fish model) which are more suitable for determing of fungal pathogencity. I suggest removing “pathogen morphotypes” from the manuscript and I want authors to state in the Material and Methods Section, that the positive growth on mammalian body temperature (37°C) was only used to select isolates which could be potential human pathogens. Also, TSA is medium more suitable for bacterial cultivation, rather the fungal. Is there any literature data for using the growth of fungi on TSA for detecting potential pathogens?

Line 288. I suggest using control buildings instead of reference building

Line 303. Why the authors chose to use passive sedimentation of air for sampling of air-borne fungi and bacteria? If using the air-samplers in the tested rooms facilities was not applicable, that should be stated in the manuscript. Also, if the passive sedimentation is used, authors should use to consider Omelyansky's formula to estimate the concentration of viabile fungal propagules in the air.

Line 352. Which identification keys were used for morphological identification of fungal isolates?

Line 357. Molecular identification of fungal isolates must be more detailed described. How DNA was isolate from fungi? Which primers were used? Primers references must be provided. Which PCR protocol was used for DNA fragments amplification? How was the PCR reaction mixture prepared? How the DNA was purified? How was the DNA sequencing conducted?

Results

I suggest authors to rewrite the result section after estimation the concentration of viabile fungal propagules in the air (CFU /m3 of air), and to change tables accordingly. Also, the photographs of fungal colonies and micrographs of reproductive fungal structures are not adequate for the Journal format. I suggest checking the Samson et al (2010) for the some good photographic illustrations of fungal colonies and microscopic structures.

Reviewer 2 Report

This study describes the total cultivable microbiota in settled 17 airborne dusts collected from 7 sport facilities in Finland. Some issues needed to be addressed:

Introduction section: Figure 1 need to be more explained: left panel is from an outdoor panel? 

Results: Regarding the sentence from line 82 and 83 it should be deleted since TSA is for bacteria cultivation (has cloranfenicol) and MEA (has antibiotic) is for fungi cultivation. Otherwise, should be mentioned the culture media composition.

In Series Versicolores should be mentioned section Nidulantes; Replace Section by section.

Line 195: Paecilomyces and Aspergillus should be in italic

Lines 207 to 222: Italics are missing

Considering the mention of the dermatophytes importance in the assessed setting the inclusion of a specific culture media for these fungi should be included

Conclusions section is missing

Reviewer 3 Report

The paper is devoted to the topic, which is important for public health – abundance and composition of culturable microorganisms (mostly fungi) in the indoor air of sport facilities in Finland. However, definitely, in the present form the paper cannot be considered for publication in a scientific journal because of its low quality.

Abstract is vaguely written, contains unnecessarily detail information on the identification of the same isolates both to the section and species level, but does not contain an important information on the overall number of fungal species isolated from the dust samples.

The methodology used seems rather strange for me. Why to previously trap the dust into Petri dishes and then to cultivate it in a complicated several-stepped way? Why not to use the well-known sedimentation method exposing the Petri dishes with a cultivation medium (MEA, for example) with antibacterial antibiotic (for the cultivation of fungi), antifungal antibiotic (for the cultivation of bacteria), and exposing the empty Petri dishes only to trap dust particles? Even in the specific equipment using to collect air samples for microbiological purposes, like six-stage viable Anderson cascade impactor, each stage contains the Petri dishes with a cultivation medium. It allows direct isolation of air-borne microorganisms avoiding their loss during the above indirect isolation and enriching their composition and abundance.

The authors should justify the use of namely neutral TSA medium at 37oC for screening the potentially pathogenic fungi (only the temperature of human body is clear) by the appropriate explanation and relevant reference(s). The authors should also justify the use of the obviously moldy hay barn in a village as a reference place. In my strong opinion, the outdoor air as a main source of the indoor dust in the sport facilities would be much more suitable as reference samples than the indoor dust in the chosen rural buildings.

Line 350: what does this awkward expression mean? The dividing isolates into morphotypes should be primarily based on the microscopic morphological characteristics: type of sporulation and size of sporulation structures and spores. Why did the authors mention ascomata and sporangia if the list of isolated species does not contain either teleomorphic ascomycetes or zygomycetes?

Lines 356-357: the methodology used for the molecular identification of species needs to be presented in a proper way.

The Result section is chaotically written and greatly repeats the content of the tables, which are mostly built and titled in a way that obscures their reading and understanding. Number of isolates (CFUs) – per what? Petri dish? Overall? Where the numbers of fungal species isolated overall and in the dust from each facility are, (it is important and informative characteristic)? As well as in the abstract, the information on the long identification process (morphotype-section (aspergilli)-species) is unnecessarily detailed.

Table 3 is rather meaningless. The pictures are of poor quality, the numbers being unrelated to concreate species give little information. The tables 3, 4, 5, and 6 can be certainly combined in a single table containing full list of the identified fungal species (without morphotypes, sections, etc.), densities of their CFUs in each sport place, and the information on their potential pathogeneity. Figure 3 as well as Figure 4 are needless because of hardly "readable" colony pictures and very complicated caption with all those arrows (fig. 3).

Overall, the location of Material and Methods section at the end of the paper (probably, according to the journal guidelines) substantially complicates the results' reading.

The Discussion is insufficient and consists mostly of the commonplaces. Lines 224-232 just repeat the results. Why did not the authors discuss the relationship of the composition and amount of air fungi with the density of dust particles in the indoor air (if they measured this parameter)? Why did not they compare the composition and amount of air fungi in different sport places and did not associated these characteristics with the number of persons visited the facilities during sampling period? Once again, the comparison between indoor mycobiota and the corresponding outdoor mycobiota would be useful and informative.

The language of the paper needs to be thoroughly checked and corrected both grammatically and especially stylistically.

All other corrections and suggestions are inserted into the PDF file of the manuscript. which is attached.     
